# Vapor-Phase of Essential Oils as a Promising Solution to Prevent *Candida* Vaginal Biofilms Caused by Antifungal Resistant Strains

**DOI:** 10.3390/healthcare10091649

**Published:** 2022-08-29

**Authors:** Liliana Fernandes, Bruna Gonçalves, Raquel Costa, Ângela Fernandes, Ana Gomes, Cristina Nogueira-Silva, Sónia Silva, Maria Elisa Rodrigues, Mariana Henriques

**Affiliations:** 1Centre of Biological Engineering, LMaS—Laboratório de Microbiologia aplicada à Saúde, Campus de Gualtar, University of Minho, 4710-057 Braga, Portugal; 2LABBELS–Associate Laboratory, University of Minho, 4710-057 Braga, Portugal; 3Aromas Aqua Spa–clínica saúde, Praça 5 Outubro nº 32, Vila Verde, 4730-731 Braga, Portugal; 4Primary Care Centre, 4710-374 Braga, Portugal; 5Family Care Unit + Carandá, Primary Care Centre, 4710-374 Braga, Portugal; 6Life and Health Sciences Research Institute, School of Medicine, University of Minho, 4710-057 Braga, Portugal; 7Life and Health Sciences Research Institute, 3B’s-PT Government Associate Laboratory, 4710-057 Braga, Portugal; 8Department of Obstetrics and Gynecology, Hospital de Braga, 4710-243 Braga, Portugal; 9National Institute for Agrarian and Veterinary Research, Vairão, 4485-655 Vila do Conde, Portugal

**Keywords:** vulvovaginal candidiasis, biofilm, antifungal resistance, alternative therapies, essential oil, vapor-phase, phytotherapeutic application

## Abstract

Background: Vulvovaginal candidiasis (VVC) is a disease with high incidence, a huge impact on the quality of life and health of women, and which represents a great challenge to treat. The growing need to apply antifungal intensive therapies have contributed to an emergence of drug-resistant *Candida* strains. Thus, effective therapeutic options, to meet the antifungal-resistance challenge and to control high resilient biofilms, are urgently needed. This study aimed to investigate the antifungal activity of essentials oils (EOs) on drug-resistant *Candida* vaginal isolates. Method: Therefore, the antimicrobial effect of tea tree, niaouli, white thyme, and cajeput EOs on the planktonic growth of *Candida* isolates was initially evaluated by an agar disc diffusion method. Then, the vapor-phase effect of tea tree EO (VP-TTEO) on biofilm formation and on pre-formed biofilms was evaluated by crystal violet staining, XTT reduction assay, colony forming units’ enumeration, and scanning electron microscopy. Results: The results revealed high antifungal activity of EOs against drug-resistant *Candida* isolates. Additionally, the VP-TTEO showed a significant inhibitory effect on the biofilm formation of all tested isolates and was able to provoke an expressive reduction in mature *Candida albicans* biofilms. Conclusions: Overall, this study suggests that the VP-EO may be a promising solution that is able to prevent biofilm-related VVC caused by antifungal-resistant strains.

## 1. Introduction

Vulvovaginal candidiasis (VVC), an extremely common clinical condition that affects millions of women every year around the world, is an infection of the female reproductive tract mucosa with vulvar and/or intravaginal changes that consequently causes physical and mental suffering, involving considerable direct and indirect costs [1]. Furthermore, complicated episodes of VVC have been suggested to enhance the risk of acquiring HIV and other complications including pelvic inflammatory disease, infertility, ectopic pregnancy, abortion, and menstrual disorders [2]. During episodes of VVC, the innate immune responses may differ, and this fact suggests that the host environment performs a key role in the host–pathogen interaction [3]. *Candida* organisms are capable of colonizing the vaginal environment asymptomatically; however, several factors may contribute to the development of infection, including pregnancy, immunosuppression (HIV/AIDS or diabetes mellitus), sexual activity, hormonal fluctuations, smoking, obesity, and the use of sodium glucose cotransporter 2 (SGLT2) inhibitors, broad spectrum antibiotics, or oral contraceptives [4,5,6,7]. Furthermore, the vaginal pathogenicity of *Candida* organisms is facilitated by several virulence factors including the biofilm formation on vaginal walls and intrauterine devices (IUDs) [8]. Biofilms show high resilience to common antifungal agents and host immune responses, requiring long and intensive therapies and, in more severe cases, the removal of the infected device in order to avoid relapses [9,10]. Of note, the increase in at-risk individuals and the widespread use of over-the-counter antifungals have been suggested to contribute to the increase/selection of drug-resistant strains [11,12]. Although *Candida albicans* is still the species that most frequently causes CVV, there is growing concern regarding the incidence of the non-*Candida albicans Candida* (NCAC) species. Among them, *Candida glabrata* or *Candida krusei* have shown low intrinsic susceptibility to common antifungals and, in addition, the ability to acquire resistance after exposure to the agent [4,13]. Therefore, the increasing incidence of drug-resistant vaginal strains, the high difficulty to treat biofilm-related VVC, and the extremely limited and toxic therapeutic options for this disease make the development of more efficient and safer strategies to control it crucial. In this context, therapies developed from natural products have emerged [14,15]. Honey, saponins, polyphenols (green tea), garlic extract, essential oils (EOs), usnic acid, and peptides have been shown to exert antimicrobial activity, both *in vivo* and *in vitro*, and, therefore, EOs stand out as the most promising [16,17,18]. EOs can be a valuable alternative, since they present low toxicity and a wide range of proprieties, including anti-*Candida* activity [19,20,21,22,23]. Nevertheless, EOs were shown to inhibit the growth of beneficial vaginal bacteria at high concentrations and may cause skin irritation, limiting their potential use in VVC [24]. Interestingly, EOs show vapor-phase (VP) antimicrobial activity, which has been shown to present higher activity towards *Candida* growth than the liquid phase, and this would avoid direct contact of the EO with the skin [25]. Currently, although several approaches in the study of planktonic status have been reported, no standard assay has been established to evaluate microbial inhibition/ inactivation by EOs in the vapor phase. Nevertheless, appropriate the management of EOs during their use as an antimicrobial agent is very important for the system used, the environment, and the package to interact [26].

Thus, this study investigated the ability of the VP of EOs (VP-EOs) to inhibit the biofilm formation and to destroy the mature biofilms of antifungal-resistant vaginal isolates of the *Candida* species; the experiment was performed on glassware only to avoid interferences. Therefore, our main aim is to contribute to the development of new, natural, and effective strategies that are able to fight or prevent VVC biofilms.

## 2. Materials and Methods

### 2.1. Essential Oils

This study evaluated the antifungal activity of four EOs, namely, tea tree (*Melaleuca alternifolia*; Florame^®^, Provence, France); niaouli (*Melaleuca quinquenervia*; Florame^®^, Provence, France); white thyme (*Thymus satureiodes;* Florame^®^, Provence, France); and cajeput (*Melaleuca cajaputii*; Florame^®^, Provence, France) EOs (all with 100% purity).

Analysis of the EOs was carried out by Florame^®^ (Provence, France), using a Gas Chromatography (GC) with Flame-Ionization Detection (FID). The major compounds of the EOs are terpinen-4-ol (43%), γ-terpinene (22%), and α-terpinene (10%) in tea tree EO, *p*-cymene, limonene, and 1,8-cineole (57%) in niaouli EO, α-terpineol, borneol (47%), and carvacrol (9%) in white thyme EO, and 1,8-cineole (60%), α-terpineol (12%), and limonene (5%) in cajeput EO. All EO samples used in this study were stored in the dark, protected by aluminum foil, at room temperature.

#### Cytotoxicity of Vapor-Phase of Essential Oils

The effect of VP-EOs (tea tree, niaouli, white thyme, and cajeput) on cell viability was evaluated by MTS ((3-[4,5-carboxymethoxyphenyl]-2-(4-sulfophenyl)-2H-tetrazolium)) test. For this, cells from human primary fibroblast cell lines (3T3-CCL 163-from the American Type Culture Collection) were first cultured in DMEM (Biochrom, Berlin, Germany) with 1% streptomycin/penicillin-containing antibiotic (Biochrom, Berlin, Germany) and 10% fetal bovine serum (FBS; Sigma-Aldrich, St Louis, MO, USA) at 37 °C and 5% CO_2_ for 24 h. Then, the cells were trypsinized and seeded in glass wells (200 µL per well) at a final concentration of 1 × 10^5^ cells/mL under the same cultivation conditions described above. After 24 h of incubation, the cell cultures were treated with VP-EOs, and 25 µL of each EO (100%) was discarded in a sterile white disk that was placed next to the wells; the set was kept inside a plate of glass for another 24 h at 37 °C and 5% CO_2_. After this time, the wells were washed twice with phosphate-buffered saline (PBS 1×), and then 80 µL of MTS (CellTiter 96 Aquoous One Solution Cell Proliferation Assay, Promega, Madison, WI, USA) with 1% phenol-free DMEM was added to each glass well for 1 h. After this time, the absorbance (OD 490 nm) was measured in a Heales MB-580 microplate reader (Heales, Shenzhen, China). The VP-EOs cytotoxicity results were expressed as the percentage of viable cells in relation to the OD 490 nm of cells cultured without VP-EOs (100% cell viability). These experiments were performed twice, and each analysis was performed in duplicate.

### 2.2. Microorganisms and Initial Culture Conditions

For this study, *C. albicans* (*n* = 3), *C. glabrata* (*n* = 8), *C. krusei* (*n* = 1), and *Candida guilliermondii* (*n* = 1) vaginal isolates belonging to a collection of yeasts created by the *Candida* Research Group of the Centre of Biological Engineering of University of Minho, in the scope of a large-scale recovery of vaginal specimens of *Candida* carried out in health centres and a university campus in the north of Portugal, were used [7]. The identity of all isolates was obtained by PCR-based sequencing using specific primers (ITS1 and ITS4) against the 5.8S subunit gene [7,27]. Minimum inhibitory concentration (MIC) for each species was determined by the Epsilometer-test (E-test) methodology and according to the guidelines provided by the manufacturer. Fernandes et al. [7] also tested reference strains with MICs characterized by the CLSI19 microdilution method to ensure the reliability of the E-test results. In addition, a potential interpretation of MICs based on CLSI M60 was also performed [7]. All the isolates used are at least resistant to one of the antifungal agents (fluconazole, ketoconazole, or caspofungin) [28,29]. Table 1 presents the MIC of each antifungal and the features reported by women at the moment of vaginal sample collection, including symptoms of vaginal infection, previous vaginal infections, use of over-the-counter antifungals, and relevant health conditions. This study followed the Data Protection Legislation and was approved by the Portuguese Health Ethical Commissions (SECVS-UM 092/2017, CES-S Norte 49/218, CESHB 151/2018, CES-USLAM 23/2018, CESHSOG 5/2018) [7].

*Candida* isolates were kept at –80 ± 2 °C in Sabouraud Dextrose Broth medium (SDB; Liofilchem, Teramo, Italy) with 20% (*v/v*) glycerol (Biochem Chemopharma, Nièvre, France). Prior to each assay, the isolates were subcultured on Sabouraud Dextrose Agar (SDA; Liofilchem) plates and incubated at 37 °C for 24 h. Then, 3-5 colonies of *Candida* isolates were inoculated onto SDB for 18 h at 37 °C under agitation (120 rev/min). After this, they were centrifuged at 5000× *g* for 10 min at 4 °C and washed twice with PBS. The supernatants were discarded, and the pellets were suspended in SDB. For the subsequent analyses, the cellular density of the pre-inocula was adjusted to 1 × 10^8^ or 1 × 10^5^ cells/mL, depending on the intended analysis, using a Neubauer haemocytometer (Marienfeld, Lauda-Königshofen, Germany).

### 2.3. Evaluation of the Antifungal Activity of Essential Oils

#### 2.3.1. Growth Inhibition Analysis

The inhibitory activity of the EOs (tea tree, niaouli, and cajeput) on the growth of the drug-resistant *Candida* isolates was evaluated using the disk-diffusion agar method [30]. Briefly, SDA plates were inoculated by using a swab dipped in cell suspensions (pre-inocula) adjusted 1 × 10^8^ cells/mL. Then, 25 µL of each EO (100%) was discarded on sterile blank disks (Liofilchem^®^) and placed atop of the plates (disks without EOs were also included as control). The SDA plates were incubated for 24 h (at 37 °C); then, the inhibition zones induced by the EOs were measured (mm).

#### 2.3.2. Effect of the Vapor-Phase of Essential Oils on Biofilms

The effect of the VP-EOs on biofilm formation and on the mature biofilms (24 h-old) of one strain of each *Candida* species (*C. glabrata*, *C. albicans*, *C. krusei*, and *C. guillermondii*) was evaluated. The strains and the EO for this assay were selected based on the results of the evaluation of the antimicrobial activity of the EOs in Section 2.3.1. Biofilms were developed as described by Stepanović et al., with some modifications due to the use of volatile compounds [31]. In order to determine the effect of tea tree VP-EO (VP-TTEO) on biofilm formation, *Candida* cellular suspensions adjusted to 1 × 10^5^ cells/mL were transferred to glass wells (1 mL per well), and 25 µL of the tea tree EO (100%) was discarded on a sterile blank disk which was placed near the wells (the set was kept inside a glass plate). Plates were incubated for 24 h at 37 °C under agitation in an orbital shaker (120 rev/min). Additionally, biofilms were pre-formed during 24 h and, after this time, were incubated in the presence of 25 µL of tea tree EO discarded on a sterile blank disk for an additional 24 h. As a control, biofilms were formed without any contact with the VP-TTEO for 24 h and 48 h.

Biofilms were analysed with the following parameters: determination of the *Candida* cultivable cells’ numbers through the colony forming units (CFUs) counting methodology, quantification of biofilm biomass by staining with crystal violet (CV), determination of metabolic activity by XTT reduction assay, and evaluation of biofilm cell morphology by scanning electron microscope (SEM) [32,33].

##### Quantification of *Candida* Biofilm Biomass

To quantify the biomass resulting from the EO assay, the biofilms treated with VP-TTEO and the respective controls were fixed with 1 mL of methanol, which was removed after 15 min and dried at room temperature. After, 1 mL of CV (1%) was added to each well and incubated for 5 min. The glass wells were then gently washed with sterile, ultra-pure water, and 1 mL of acetic acid (33%) was added to release and dissolve the stain. Thus, 200 µL of the solution obtained from each glass well was immediately transferred to a microtiter plate, and the absorbance of each condition was read at 570 nm in triplicate on a Multiskan™ FC microtiter plate reader (Thermo Scientific™, Waltham, MA, EUA) [33].

##### Quantification of *Candida* Biofilm Cultivable Cells

The number of cultivable cells in the biofilms was estimated using the CFU counting methodology. Briefly, the biofilms were washed with PBS to remove non-adherent cells and were then scraped from the wells with 1 mL of PBS. The suspensions obtained were serially diluted to 10^8^ in PBS and then plated on SDA. SDA plates were incubated (24 h at 37 °C), and the number of colonies grown was counted and translated into colony forming units per milliliter (Log (CFU/mL)) [32].

##### Quantification of Metabolic Activity of *Candida* Biofilms Cells

An XTT reduction assay was used to determine the biofilm metabolic activity after contact with VP-TTEO. Therefore, the culture medium was aspirated after 24 h or 48 h, and non-adherent cells were removed by washing with PBS. Then, 200 μL of a solution containing 100 μg/μL of XTT (2,3-(2-methoxy-4-nitro-5-sulphophenyl)-5-[(phenylamino)carbonyl]-2H-tetrazolium ydroxide) (Sigma–Aldrich) and 10 μg/μL of phenazine methosulphate (PMS) (Sigma–Aldrich) were added to each well and incubated at 37 °C (120 rev/min) for 3 h, in the dark, and protected by aluminium foil. Thus, 150 µL was transferred from each glass well to a microtiter plate, and the colorimetric changes were measured at 490 nm using a Heales MB-580 microtiter plate reader [32,34].

##### Scanning Electron Microscopy (SEM)

To examine the morphology of the biofilm cells after contact with VP-EOs, the biofilms treated with VP-TTEO and the respective controls were analyzed by SEM. For this, biofilms formed on glass coupons, under the same conditions as those described above, for 24 h, were dehydrated with ethanol (using 70% ethanol for 10 min, 95% ethanol for 10 min and 100% ethanol for 20 min). The samples were kept in a desiccator for at least 48 h. Prior to observation, coupons were sputtered with gold and observed with Phenom Desktop SEM (Thermo Scientific™) [32].

### 2.4. Statistical Analysis

The Prism software package (GraphPad Prism version 6.01 for Windows, GraphPad Software, San Diego, CA, USA) was used to perform the statistical analysis of the results obtained in this study. For that, the cell cultivability of biofilms treated with VP-TTEO was compared with that of untreated biofilms using one-way ANOVA and Tukey’s multiple comparisons test (confidence level of 95% and statistical significance was assumed at *p* < 0.05). For all assays, three independent experiments were carried out (independent pre-inocula), and each analysis was performed in duplicate.

## 3. Results

### 3.1. Cytotoxic Effect of Essential Oils

The cytotoxic effects of the tea tree, niaouli, cajeput, and white thyme VP-EOs were evaluated (Figure 1). Viability of the unexposed to VP-EOs cell cultures (control) was set at 100% to compare with the responses of the cell cultures exposed to VP-EOs. In cultures exposed to the VP-EOs of the tea tree, cajeput, and niaouli, the viability was greater than 70%. However, the viability of cultures incubated with the VP of white thyme EOs was less than 70%.

### 3.2. Antifungal Activity of Essential Oils

The inhibitory activity of the EOs (tea tree, niaouli and cajeput) on the growth of the antifungal-resistant isolates of *C. albicans*, *C. glabrata*, *C. guilliermondii*, and *C. krusei* was evaluated using the disk-diffusion agar method, and the results are summarized in Table 1. The tea tree EO had the greatest inhibitory effect (20–39 mm of inhibition zone), followed by the cajeput (10–21 mm) and niaouli (10–17 mm) EOs (Table 2). Among the isolates, the highest inhibition was found in *C. glabrata* Cg1, and the lowest was found in *C. krusei* Ck1 (Table 2).

### 3.3. Antifungal Activity of the Vapor-Phase of Essential Oils on Biofilms Formation

The effect of the VP-TTEO on biofilm formation and on the mature biofilms (24 h-old) of *C. albicans* Ca2, *C. guillermondii* Cgi 1, *C. glabrata* Cg7, and *C. krusei* Ck1 was evaluated. The VP-TTEO resulted in a significant reduction of two to four orders of magnitude (Log CFU/mL) in the cell cultivability of the biofilm in all the isolates tested (Figure 2A). Indeed, VP-TTEO reduced 2 Log CFU/mL in *C. albicans* Ca2 and *C. krusei* Ck1, 2.5 Log CFU/mL in *C. glabrata* Cg7, and the greatest effect of 4 Log CFU/mL was observed in *C.*
*guillermondii* Cgi1. In relation to biomass quantification, a statistically lower amount of biomass was found on the biofilm formation of *C. albicans* Ca2 (*p*-value < 0.0001) and *C. guilliermondii* Cgi1 (*p*-value < 0.1) (Figure 2B). Regarding the metabolic activity in the biofilm formation, it was observed that this parameter decreased for all species tested, mainly in *C. glabrata* Cg7 (*p*-value < 0.01) and *C. guilliermondii* Cgi1 (*p*-value < 0.0001) (Figure 2C). The SEM images (Figure 2D) show evident alterations in the morphology of the biofilm cells after contact with VP-TTEO, and the main changes observed are membrane damage and alteration in hyphal capacity.

### 3.4. Antifungal Activity of the Vapor-Phase of Essential Oils on Mature Biofilms

VP-TTEO significantly reduced (4 Log CFU/mL) the number of viable cells of *C. albicans* Ca2 (Figure 3A) in mature biofilms (24 h-old). Concerning the quantification of biomass, no significant differences were observed between the VP-TTEO and the control (Figure 3B). Instead, there was a significant decrease in metabolic activity in *C. guilliermondii* Cgi1 (*p* < 0.1) and a slight decrease in metabolic activity in *C. krusei* Ck1 and *C. glabrata* Cg7 after contact with VP-TTEO. From the SEM images (Figure 3D) of the treatment with VP-TTEO, alterations in the cell morphology of the *C. albicans* Ca2 and *C. glabrata* Cg7 biofilms are detected. In fact, *C. albicans* Ca2 changed from yeast to hypha after exposure to VP-TTEO.

## 4. Discussion

This study evaluated the antifungal activity of EOs and their VP against antifungal-resistant vaginal isolates of various *Candida* species. Approximately 50% of the isolates were collected from women with symptoms of vaginal infection (with VVC) and the remaining were from asymptomatic women (colonized with *Candida* organisms) (Table 1). Importantly, almost all women reported one or more vaginal infections prior to the collection of the drug-resistant strains, suggesting that previous antifungal treatments may have contributed to the acquisition of resistance [11,12]. Additionally, the use of over-the-counter antifungals to treat self-diagnosed VVC, which was reported by 30% of the women of this study (Table 1), can also be suggested to contribute to the selection of low-susceptible strains [29,35]. Moreover, some women reported IUD use, pregnancy, or immunocompromised conditions (Table 1); therefore, these women are at greater risk of developing VVC and, consequently, are more likely to fail treatment [36,37]. Of note, all clinical vaginal isolates used during this study are at least resistant to one of antifungal agents (fluconazole, ketoconazole, and caspofungin) (Table 1). Indeed, the extremely limited options to treat VVC caused by drug-resistant strains make the development of new strategies designed to meet the drug-resistance challenge crucial [38]. EOs appear as a potential alternative, presenting low toxicity and the main advantage of their natural antimicrobial agents’ ability to avoid the development of antifungal resistance [39,40,41]. The cytotoxic effect of the tea tree, niaouli, cajeput, and white thyme VP-EOs was determined against the 3T3 cell line (Figure 1). The results (Figure 1) show that the VP of the tea tree, cajeput, and niaouli EOs were not cytotoxic, since the relative cell viability is lower than 70% of the control (no EOs), based on ISO 10993-5:2009 [42]. So, an initial screening of the non-cytotoxic EOs in test was performed for the antifungal activity of the tea tree, cajeput, and niaouli EOs against antifungal-resistant isolates of *C. albicans, C. glabrata, C. guilliermondi*, and *C. krusei*, the results of which were evaluated. The results revealed that all EOs were able to inhibit the growth of the tested isolates, although with different impacts (Table 2).

Among the isolates, the highest inhibition was found in *C. glabrata* Cg1 and the lowest in *C. krusei* Ck1 (Table 2). Previous studies have also reported the high antifungal activity of tea tree EOs against susceptible and drug-resistant strains of various *Candida* species, including *C. albicans*, *C. glabrata*, and *C. krusei* [43,44,45]. In fact, the tea tree EO was found to increase yeast cell permeability and membrane fluidity and to inhibit the acidification of the medium [46]. Moreover, Keereedach, Hrimpeng, and Boonbumrung recently demonstrated that the cajeput EO has an antifungal effect on fluconazole-resistant *C. albicans* isolates [47]. These investigators showed that the cajeput EO reduced the expression level of the *C. albicans MDR1* gene, which encodes the multidrug resistance protein 1 [47]. Of note, the antifungal activity of EOs has been shown to be influenced by several factors, such as the original species, geographic and climatic conditions, the biological and physical–chemical properties of the soil, and storage conditions [48]. As such, an adequate control of these factors may allow for the maximization of the antifungal activity of EOs.

One of the most important virulence factors of the *Candida* species is their ability to form biofilms, which promote the development of VVC and make its treatment extremely difficult and often ineffective [49]. Therefore, a more effective, low-toxic, and inexpensive solution to treat biofilm-related VVC is urgently needed. Eos have been suggested as a promising solution due their ability to inhibit the biofilm formation and reduce pre-formed biofilms of various *Candida* species, including *C. albicans*, *C. glabrata*, *C. parapsilosis*, and *C. krusei* [50]. Furthermore, EOs present high volatility, and their VP was shown to possess higher antimicrobial activity than the liquid phase [25,51]. Inouye et al. suggested that, in their aqueous state, EOs’ lipophilic molecules associate to form micelles and thus suppress their binding to organisms, while those in the VP allow for free binding [52]. As such, this study investigated, for the first time, the effect of the VP-EO on *Candida* biofilm formation and on pre-formed biofilms of antifungal-resistant strains (*C. albicans* Ca2, *C. guilliermondii* Cgi1, *C. glabrata* Cg7, and *C. krusei* Ck1). Importantly, *C. albicans* Ca2 and *C. krusei* Ck1 present an extremely high resistance to fluconazole; the most prescribed antifungal agent, *C. glabrata* Cg7, is resistant to ketoconazole, a less common azole; and *C. guilliermondii* Cgi1 presents resistance to caspofungin, a highly toxic antifungal (Table 1). The tea tree EO was selected for the assays with biofilms due their high antifungal activity against these drug-resistant isolates (Table 2).

In order to study the effect of VP-TTEO on the biofilm formation (Figure 2) and pre-formed biofilms (Figure 3) of antifungal-resistant *Candida* isolates (*C. albicans* Ca2, *C. guilliermondii* Cgi1, *C. glabrata* Cg7, and *C. krusei* Ck1), the number of cultivable cells (Figure 2A and Figure 3A), the total biomass (Figure 2B and Figure 3B), metabolic activity (Figure 2C and Figure 3C), and cells’ morphology (Figure 2D and Figure 3D) were evaluated.

The results of the cultivable cells’ number revealed that the tea tree EO had a high inhibitory effect on the biofilm formation of all the isolates tested (Figure 2A). The highest inhibition was obtained in the *C. guilliermondii* Cgi1 biofilms, followed by those of *C. glabrata* Cg7 (2.5–4 Log CFU/mL). In addition, the biomass quantification revealed a significant alteration of two species, *C. albicans* Ca2 and *C. guilliermondii* Cgi1 (Figure 2B). Although CV staining quantifies both dead and live cells and the biofilm matrix, it is possible that, while the biofilm was not completely removed, a large portion of the cells were dead [53]. The effect of VP-TTEO on the metabolic activity of biofilm formation cells was also evaluated (Figure 2C), and a decrease was observed in the four *Candida* species tested, with greater emphasis on *C. guilliermondii* Cgi1, followed by *C. glabrata* Cg7. In agreement with the results in Figure 2A and Figure 2B, *C. guilliermondii* Cgi1 showed a greater decrease in metabolic activity. The results of these three parameters are confirmed by the SEM images (Figure 2D). In fact, when comparing the image of the untreated biofilm cells (control) with the images after contact with VP-TTEO, an increase in the number of damaged cells (membrane change, no hyphae or burst cells) is evident for the four *Candida* strains tested.

Additionally, the VP-TTEOs were also able to reduce the number of viable cells on the mature biofilms (24h-old) of *C. albicans* Ca2 (Figure 3A) and to change their morphology (Figure 3D). Indeed, due to cellular stress, the *C. albicans* can transition from yeast to hypha [54]. The results obtained (Figure 3C) on the metabolic activity of mature biofilm’s cells show a significant reduction in the metabolic activity in *C. guilliermondii* Cgi1 and a slight decrease in the metabolic activity of the *C. krusei* Ck1 and *C. glabrata* Cg7 biofilms’ cells cultivated in the presence of VP-TTEO, with morphology alteration of the *C. glabrata* biofilm cells, compared to the absence of the EOs. Interestingly, in the *C. albicans* strain and the *C. guilliermondii* strain, there is an opposite effect between metabolic activity and the biomass quantification, both in the biofilm formation and in mature biofilms; this curious fact may be due to the unique structure of each biofilm, and the biomass quantification assay did not differentiate between living and dead cells. These results, together with those obtained with CV (Figure 2B), the cultivable cells’ number (Figure 2A), and SEM (Figure 2D) suggest a more promising effect of VP-TTEO on biofilm formation.

Overall, the results show a significant inhibitory effect of the VP-TTEO on the development of *Candida* biofilms of antifungal-resistant isolates. However, the induced effect by VP-TTEO may differ according to the *Candida* species due to the differences that each species presents, both in its planktonic state and in the biofilm structure. In this sense, it would be important to determine the mode of action of VP-TTEO on different *Candida* species. Furthermore, a possible factor that can influence this effect is the distance of the EO from *Candida* cells and the atmospheric dimension of the VP-EO. Therefore, in future studies, it would be important to evaluate these factors together with *in vivo* and clinical trials.

## 5. Conclusions

VVC has been considered a public health problem due to its serious negative consequences, high incidence, and increasing difficulty to treat. The emergent resistance of *Candida* organisms to conventional antifungal agents have been pushed the development of more effective strategies to combat them. This study suggests a promising use of VP-EOs to prevent biofilm-related VVC caused by drug-resistant strains. Interestingly, a VP-mediated treatment has additional advantages, including high efficacy without requiring direct contact of the liquid phase with the skin and an easy mode of application (for instance on women’s underwear). Of note, some antifungal-resistant isolates used in this study were collected from women (symptomatic or not) presenting clinically relevant conditions such as pregnancy and immunodepression, in which the prevention and quick treatment of VVC are imperative. However, the prophylactic treatment of VVC with common antifungals, besides being not always effective, may contribute to the selection of strains with low antifungal susceptibility. In contrast, the VP-EOs seem to be a safe and effective solution to prevent biofilm-related VVC in at-risk women such those using IUDs and those who are immunosuppressed and pregnant. In this particular study, tea tree, cajeput, and niaouli EOs present important high antifungal activity, and VP-TTEO presents an important preventive effect in biofilm-related VVC caused by antifungal-resistant strains. Importantly, *Candida* organisms have the ability to form multi-species biofilms, either of the *Candida*–*Candida* species as well as of *Candida*–bacteria, and, thus, future studies to ascertain the efficacy of VP-EOs on multi-species biofilms will be also of value. Additionally, further studies are needed to explain the mechanisms of action of EOs and the possible toxicity associated with the VP of these compounds to establish whether they can be safely used as antifungal agents for therapy or prophylactic treatment for biofilm-related CVV.

## Figures and Tables

**Figure 1 healthcare-10-01649-f001:**
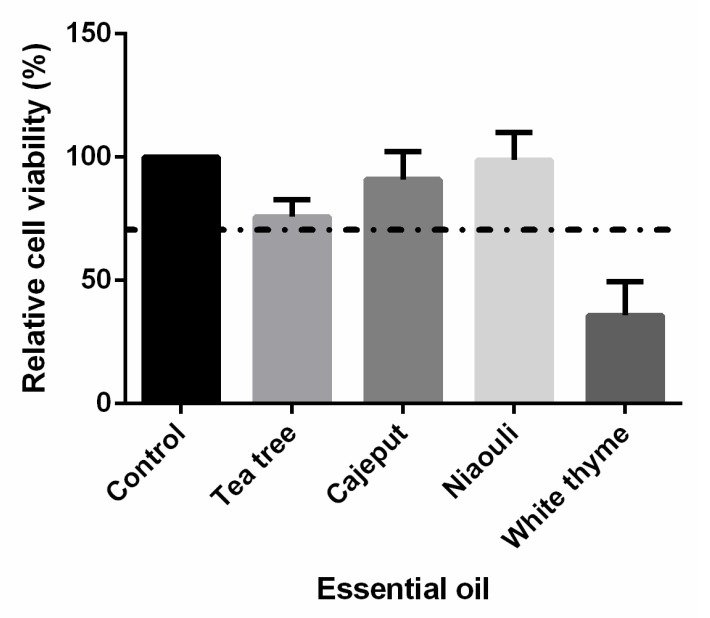
Vapor phase of essential oils’ (VP-EOs) cytotoxicity, expressed as the percentage of viable cells in relation to the absorbance values (OD 490 nm) of cells cultured without VP-EOs (100% cell viability). The dashed line represents the normative limit of 70% metabolic activity.

**Figure 2 healthcare-10-01649-f002:**
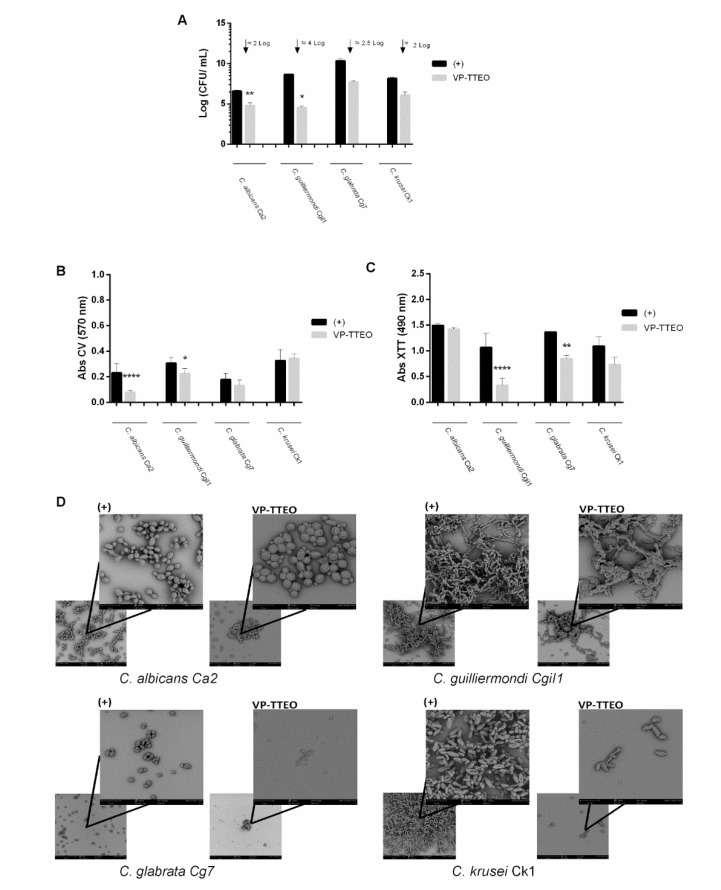
Effect of the vapor phase of tea tree essential oil (VP-TTEO) on the biofilm formation of antifungal-resistant *Candida* isolates. Biofilms of *C. albicans* Ca2, *C. guilliermondii* Cgi1, *C. glabrata* Cg7, and *C. krusei* Ck1 were developed in the absence (control) and presence of the VP-EOs: (**A**) Number of cultivable cells (Log CFU/mL). (**B**) Absorbance values of crystal violet solutions (Abs CV) and (**C**) absorbance values of XTT solutions (Abs XTT). * indicates statistical reduction in biofilms cell cultivability in comparison with the respective control (* *p* < 0.1, ** *p* < 0.01, **** *p* < 0.0001). (**D**) Scanning electron microscope (SEM) images. The bar represents 30 µm (lower magnification image) or 10 µm (higher magnification image).

**Figure 3 healthcare-10-01649-f003:**
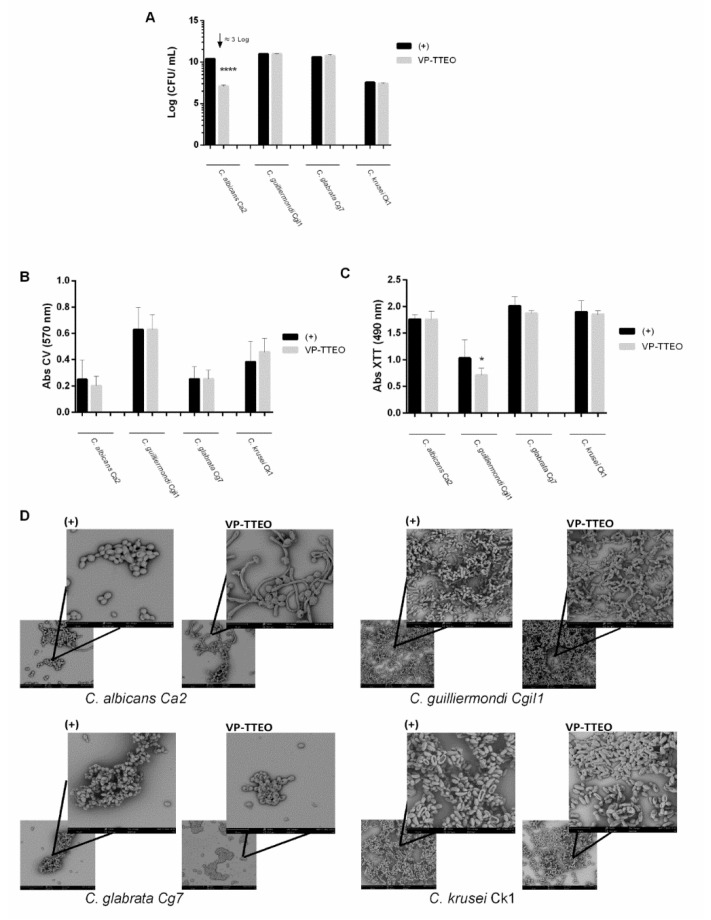
Effect of the vapor phase of tea tree essential oil (VP-TTEO) on pre-formed biofilms of antifungal-resistant *Candida* isolates. Biofilms of *C. albicans* Ca2, *C. guilliermondii* Cgi1, *C. glabrata* Cg7, and *C. krusei* Ck1 were developed in the absence (control) and presence of the VP-EOs: (**A**) Number of cultivable cells (Log CFU/mL). (**B**) Absorbance values of crystal violet solutions (Abs CV) and (**C**) absorbance values of XTT solutions (Abs XTT). * indicates statistical reduction of biofilms cell cultivability in comparison with the respective control (* *p* < 0.1, **** *p* < 0.0001). (**D**) Scanning electron microscope (SEM) images. The bar represents 30 µm (lower magnification image) or 10 µm (higher magnification image).

**Table 1 healthcare-10-01649-t001:** List of vaginal isolates used in this study and respective data, including species, features of the women at the moment of sample collection, and minimum inhibitory concentration (MIC) of fluconazole, ketoconazole and caspofungin for each isolate.

Species	Isolate	Women’s Features				MIC (µg/mL)		
		Symptoms of Infection	Previous Infections	Use of over-the-Counter Antifungals	Relevant Conditions	Fluconazole	Ketoconazole	Caspofungin
** *C. glabrata* **	**Cg1**		Yes	Yes		96 (R)	2 (R)	0.25 (I)
	**Cg2**		Yes	Yes		64 (R)	2 (R)	0.19 (I)
	**Cg3**		Yes		Diabetes	16 (SDD)	1.5 (R)	0.19 (I)
	**Cg4**		Yes			4 (SDD)	0.032 (S)	0.5 (R)
	**Cg5**		Yes	Yes		8 (SDD)	1.5 (R)	0.064 (S)
	**Cg6**	Yes	Yes			>256 (R)	1.5 (R)	0.032 (S)
	**Cg7**		Yes		Cancer	16 (SDD)	1.5 (R)	0.064 (S)
	**Cg8**	Yes	Yes		IUD	>256 (R)	6 (R)	0.047 (S)
** *C. albicans* **	**Ca1**	Yes	Yes	Yes		1 (S)	0.032 (S)	1 (R)
	**Ca2**	Yes	Yes		Pregnancy	64 (R)	32 (R)	0.064 (S)
	**Ca3**	Yes	Yes		Auto-immune disease	>256 (R)	0.38 (S)	0.19 (S)
** *C. krusei* **	**Ck1**	Yes	Yes			64 (R)	3(R)	0.38 (I)
** *C. guilliermondii* **	**Cgi1**					8	0.023	32 (R)

**Table 2 healthcare-10-01649-t002:** Anti-*Candida* activity of essential oils on drug-resistant isolates evaluated through the disk-diffusion method.

Species	Isolate	Essential Oil Inhibition Zone (mm)		
		Tea tree	Cajeput	Niaouli
** *C. glabrata* **	**Cg1**	39.0 ± 1.7	21.0 ± 2.6	13.0 ± 2.0
	**Cg2**	29.7 ± 0.6	18.8 ± 1.5	12.7 ± 1.2
	**Cg3**	28.0 ± 0.0	10.3 ± 0.5	10.4 ± 0.9
	**Cg4**	34.0 ± 1.0	14.8 ± 0.5	12.8 ± 0.5
	**Cg5**	26.3 ± 4.9	19.3 ± 3.8	16.0 ± 1.0
	**Cg6**	32.8 ± 3.2	19.0 ± 1.7	15.8 ± 2.2
	**Cg7**	20.5 ± 1.0	10.5 ± 1.0	10.5 ± 1.0
	**Cg8**	20.0 ± 0.8	10.3 ± 0.5	10.0 ± 0.0
** *C. albicans* **	**Ca1**	22.5 ± 2.4	25.5 ± 3.5	10.8 ± 1.0
	**Ca2**	24.8 ± 2.4	21.5 ± 1.3	12.8 ± 1.3
	**Ca3**	23.8 ± 0.5	18.7 ± 1.5	10.8 ± 0.5
** *C. krusei* **	**Ck1**	21.8 ± 2.1	12.3 ± 0.6	12.0 ± 1.3
** *C. guilliermondii* **	**Cgi1**	28.0 ± 8.9	20.3 ± 2.5	17.3 ± 2.9

## Data Availability

Not applicable.

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
