# Peer review of "Vapor-Phase of Essential Oils as a Promising Solution to Prevent Candida Vaginal Biofilms Caused by Antifungal Resistant Strains"

_healthcare, 2022, doi:10.3390/healthcare10091649_

Round 1

Reviewer 1 Report

1.      Introduction part should include the recent study on predominant causative agents of VVC. 

2.      Ethics details must be provided.

3.      Why Candida parapsilosis was not included in the study?

4.      What criteria were used for interpretation of MIC, especially ketoconazole as shown in table 1? Author may add suitable references used for the interpretation of MIC.

5.      No reference is cited for 2.3.2.1. Quantification of Candida biofilm biomass, 2.3.2.2. Quantification of Candida biofilm cultivable cells and 2.3.2.3. Quantification of metabolic activity of Candida biofilms cells.

6.      A totally unrelated reference number 31 is cited for 2.3.2.4. Scanning electron microscopy (SEM).

7.      There are two Figure 1 in the article - one is on page no. 3 and another is on page no. 5. It seems there is a typographic error in page no. 5. It is Figure 2.  

8.      In 3.4 section on page no. 6 Figure 3 is described. Figure 3 is missing in the article.

9.      Why Cg7 and Ca2 strains out of 8 different strains of C. glabrata and 3 different strains of C. albicans respectively were selected for antifungal activity of the vapor-phase of essential oils on biofilms formation?

10.  In figure 1B max reduction was observed in C. albicans strain but in figure 1C max reduction was observed in C. guilliermondii strain. Similar observation is there in figure 2 also. How do you explain this? The probable explanation may be added in the discussion part.

11.  Data labels in figure 1A, 1B, 1C, 2A, 2B and 2C may be added.

12.  In Figure 1 and 2 – Either keep uniform bar as 10 μm or 30 μm; or mention it which one is 10 μm and which one is 30 μm

13.  Authors had demonstrated that the VP-TTEO has an effect on Candida spp. and this effect may be variable. Authors may add the probable factors determining this variable effect of VP-TTEO.

14.  In Figure 2 D SEM image of C. albicans - VP-TTEO showed more hyphae than control image. Author should add this observation in the result section and should include appropriate discussion on it.

15.  Authors should include limitations of the study e.g. clinical trials are needed before starting VP-TTEO for clinical use.

16.  References must be written as per journal’s instruction e.g reference number 31 page number 569-578 is missing

17.  In Reference species name should not start with capital letter e.g. Albicans should be written as albicans

Author Response

The authors are grateful for the comments and suggestions of the reviewer and revised the manuscript carefully. Below, a detailed explanation is provided to reviewer’s comments, enumerating the changes included in the revised version of the manuscript.

Revised portions of the manuscript are highlighted in 'Revised Manuscript'.

REVIEWER #1

Point 1: Introduction part should include the recent study on predominant causative agents of VVC. 

Response 1: The authors agree with the reviewer´s comment regarding the Introduction and, therefore, recent studies on the predominant causative agents of VVC were included in the manuscript. (Page 2, Lines 50-53).

Point 2: Ethics details must be provided.

Response 2: The required information was added to the manuscript (Page 3, Lines 150-153).

Point 3:   Why Candida parapsilosis was not included in the study?

Response 3: Candida species in this study were selected according to a study carried out by Fernandes et al. (2022) ("Vulvovaginal Candidiasis and Asymptomatic Vaginal Colonization in Portugal: Epidemiology, Risk Factors and Antifungal Pattern") and according to the susceptibility profile of each species.

Point 4:   What criteria were used for interpretation of MIC, especially ketoconazole as shown in table 1? Author may add suitable references used for the interpretation of MIC.

Response 4: The required information about the interpretation of MIC was added to the manuscript (Page 3, Lines 141-146).

Point 5:     No reference is cited for 2.3.2.1. Quantification of Candida biofilm biomass, 2.3.2.2. Quantification of Candida biofilm cultivable cells and 2.3.2.3. Quantification of metabolic activity of Candida biofilms cells.

Response 5: The correction required was made in the manuscript.

Point 6:     A totally unrelated reference number 31 is cited for 2.3.2.4. Scanning electron microscopy (SEM).

Response 6: The correction was made in the manuscript.

Point 7:     There are two Figure 1 in the article - one is on page no. 3 and another is on page no. 5. It seems there is a typographic error in page no. 5. It is Figure 2.

Response 7: The corrections were made in the manuscript.

Point 8:     In 3.4 section on page no. 6 Figure 3 is described. Figure 3 is missing in the article.

Response 8: The corrections were made in the manuscript.

 Point 9:       Why Cg7 and Ca2 strains out of 8 different strains of C. glabrata and 3 different strains of C. albicans respectively were selected for antifungal activity of the vapor-phase of essential oils on biofilms formation?

Response 9: As described in the manuscript, the selection was made for each species (C. albicans Ca2, C. guillermondiiCgil 1, C. glabrata Cg7 and C. krusei Ck1), since "C. albicans Ca2 and C. krusei Ck1 present an extremely high resistance to fluconazole, the most prescribed antifungal agent, C. glabrata Cg7 is resistant to ketoconazole, a less common azole, and C. guilliermondi Cgi present resistance to caspofungin, a highly toxic antifungal" (Page 9, Lines 516-519).

Point 10:       In figure 1B max reduction was observed in C. albicans strain but in figure 1C max reduction was observed in C. guilliermondii strain. Similar observation is there in figure 2 also. How do you explain this? The probable explanation may be added in the discussion part.

Response 10: The authors understand the reviewer's comment and already add a probable explanation in the manuscript “inC. albicans strain and C. guilliermondii strain there is an opposite effect between metabolic activity and the biomass quantification both in the biofilm formation and in mature biofilms, this curious fact may be due to the unique structure of each biofilm and the biomass quantification assay did not differentiate between living and dead cells” (Page 9, Lines 549-553).

Point 11:       Data labels in figure 1A, 1B, 1C, 2A, 2B and 2C may be added.

Response 11: The authors did not understand the comment. For the authors, the mentioned figures are completely labelled.

Point 12:    In Figure 1 and 2 – Either keep uniform bar as 10 μm or 30 μm; or mention it which one is 10 μm and which one is 30 μm

Response 12: The information was clarified in the manuscript (figures legend).

Point 13:       Authors had demonstrated that the VP-TTEO has an effect on Candida spp. and this effect may be variable. Authors may add the probable factors determining this variable effect of VP-TTEO.

Response 13: The information required was add in the manuscript (Page 9-10, Lines 557-572).

Point 14:      In Figure 2 D SEM image of C. albicans - VP-TTEO showed more hyphae than control image. Author should add this observation in the result section and should include appropriate discussion on it.

Response 14: The information required was add in the manuscript. (Page 6, Lines 401-402/ Page 9, Lines 544-545).

Point 15:        Authors should include limitations of the study e.g. clinical trials are needed before starting VP-TTEO for clinical use.

Response 15: The information required was add in the manuscript (Page 9-10, Lines 557-572).

Point 16:       References must be written as per journal’s instruction e.g reference number 31 page number 569-578 is missing

Response 16: The reference section of the manuscript was carefully revised and corrected.

Point 17:       In Reference species name should not start with capital letter e.g. Albicans should be written as albicans

Response 17: The reference species name was carefully revised and corrected.

Reviewer 2 Report

Overall, the manuscript is interesting, has good readability, and presents essential data sets for characterizing essential oils in vapor phase VP of EOs. The study is carried out in a very exhaustive way with the application of various techniques, from the characterization of essential oils to the evaluation of the antifungal activity of essential oils.

This study would contribute to the development of new forms to treat vulvovaginal candidiasis. This is an interesting, comprehensive and well-designed document, and I think it deserves publication.

It seems to me that it is necessary to attach as annexes, the results of the preliminary experiments, in order to substantiate all the work.

Author Response

The authors are grateful for the comments and suggestions of the reviewer and revised the manuscript carefully. Below, a detailed explanation is provided to reviewer’s comments, enumerating the changes included in the revised version of the manuscript.

Revised portions of the manuscript are highlighted in 'Revised Manuscript'.

REVIEWER #2

Overall, the manuscript is interesting, has good readability, and presents essential data sets for characterizing essential oils in vapor phase VP of EOs. The study is carried out in a very exhaustive way with the application of various techniques, from the characterization of essential oils to the evaluation of the antifungal activity of essential oils.

This study would contribute to the development of new forms to treat vulvovaginal candidiasis. This is an interesting, comprehensive and well-designed document, and I think it deserves publication.

It seems to me that it is necessary to attach as annexes, the results of the preliminary experiments, in order to substantiate all the work.

Response: The authors are grateful for the positive comment and the reviewer's suggestion. Nonetheless, the results of research and MICs determination from clinical isolates of Candida spp are detailed in "Vulvovaginal Candidiasis and Asymptomatic Vaginal Colonization in Portugal: Epidemiology, Risk Factors and Antifungal Pattern" (reference number 7: Fernandes, Â.; Azevedo, N.; Valente, A.; Dias, M.; Gomes, A.; Nogueira-Silva, C.; Henriques, M.; Silva, S.; Gonçalves, B. Vulvovaginal Candidiasis and Asymptomatic Vaginal Colonization in Portugal: Epidemiology, Risk Factors and Antifungal Pattern. Med Mycol 2022, 60, doi:10.1093/MMY/MYAC029.)

Reviewer 3 Report

The manuscript by Fernandez et.al. describing the effectiveness of certain natural plant extracts that are of hydrophobic in nature, is essentially is a highly relevant topic. While the methods were adequately described and data were clearly presented, this reviewer noted very many mistakes in the scientific representations of units such as ml for mL and ul for uL etc. additionally, English language and style requires thorough revision.
Once the above changes are made, this manuscript can be considered for publication.

additional point/suggestion: the authors could consider conducting an AFM based studio to ascertain the film thickness of the candida film formations and subsequent changes in thickness (if any) as a factor of different plant extract treatment of surfaces.  

Author Response

The authors are grateful for the comments and suggestions of the reviewer and revised the manuscript carefully. Below, a detailed explanation is provided to reviewer’s comments, enumerating the changes included in the revised version of the manuscript.

Revised portions of the manuscript are highlighted in 'Revised Manuscript'.

REVIEWER #3

The manuscript by Fernandes et.al. describing the effectiveness of certain natural plant extracts that are of hydrophobic in nature, is essentially is a highly relevant topic. While the methods were adequately described and data were clearly presented, this reviewer noted very many mistakes in the scientific representations of units such as ml for mL and ul for uL etc. additionally, English language and style requires thorough revision.

Once the above changes are made, this manuscript can be considered for publication additional point/suggestion: the authors could consider conducting an AFM based studio to ascertain the film thickness of the candida film formations and subsequent changes in thickness (if any) as a factor of different plant extract treatment of surfaces.  

Response: The authors are grateful for the reviewer's positive comment and suggestion, which will be considered in future works. Required corrections have been made throughout the manuscript.